# Folic acid supplementation ameliorates long-term lipid metabolism following intrauterine growth restriction

**Laiyi Zhou** [1], **Weiyun Shen** [2], **Can Liang** [1], **Qingyi Dong** [1], **Jingwei Wang** [1], **Xiaori He** [1]*

**1** Department of Neonatology, Second Xiangya Hospital, Central South University, Changsha, Hunan Province, China, **2** Department of Anesthesiology, Second Xiangya Hospital, Central South University, Changsha, Hunan Province, China

☯ These authors contributed to the work equally and should be regarded as co-first authors.
* hexiaori@csu.edu.cn

## Abstract

### Background

This study emphasizes the potential of folic acid (FA) supplementation in controlling intrauterine growth retardation (IUGR)-induced metabolic problems and improving the long-term health consequences of afflicted individuals.

### Objectives

This study investigates the effect of IUGR on long-term lipid metabolism in the liver, and the role of early-life FA supplementation on IUGR-induced metabolic dysfunctions. We aim to provide novel insights for early interventions to prevent the development of lipid metabolism abnormalities in adulthood.

### Methods

The IUGR model was induced by feeding pregnant rats a 10% low protein diet. After birth, lactating mothers were provided with a 21% normal protein diet. Offspring were initially breastfed and assigned to either an FA-supplemented diet or a standard diet without FA. Serum levels of total free fatty acids and triglycerides were measured by ELISA. Liver tissues were harvested for Hematoxylin and eosin (HE) to observe the liver structure. The expression of PPARα, Acox1, Acox3 and CPT1 in the liver tissues was analyzed using Quantitative reverse transcription-polymerase chain reaction (RT-qPCR) and Western Blot (WB).

### Results

Levels of serum triglycerides (TG) and free fatty acids (FFA) were significantly higher in IUGR rats compared to control rats from the early stages and remained elevated through to 90 days of age. The HE results displayed an irregular arrangement of

**Data availability statement:** All relevant data are within the manuscript and its Supporting Information files.

**Funding:** Ausnutria Food and Nutrition Science Research Fund（AU-YJY-B-LX-20-022）The funders had no role in study design, data collection and analysis, decision to publish, or preparation of the manuscript.

**Competing interests:** The authors have declared that no competing interests exist.

**Abbreviations:** Acox1, acyl-CoA oxidase 1; Acox3, acyl-CoA oxidase 3; Cpt1, Carnitine palmitoyl transferase 1; FA, folic acid; FAO, fatty acid oxidation; FFA, free fatty acids; IUGR, intrauterine growth retardation; PPARα, peroxisome proliferator-activated receptor α; TG, triglycerides.

hepatocytes at birth in IUGR rats, and hepatocellular vacuoles by 90 days of age in the IUGR rats. The level of lipid-related genes was significantly low in the IUGR group, including PPARα, Acox1, Acox3 and CPT1. Early postnatal folic acid supplementation significantly decreased serum levels of TG and FFA, ameliorated the pathological changes in the livers and reversed the expression of lipid-related genes.

## Conclusions

IUGR individuals are predisposed to lipid metabolism abnormalities in adulthood, which can be ameliorated by postnatal supplementation of FA, potentially serving as a therapeutic method.

## Introduction

Intrauterine growth restriction (IUGR) is a condition in which the fetus fails to achieve its intended growth potential, due to the lack of nutrients and oxygen by either maternal nutrient restriction or alterations in the placenta [1]. The global incidence of IUGR is approximately 15%, affecting around 30 million infants annually [2]. The advances in reproductive technologies and neonatal care have increased the survival of infants with IUGR into adulthood [3]. However, IUGR not only increases the risk of early mortality but also leads to long-term adverse health outcomes, including a higher risk of chronic diseases such as obesity, diabetes, and cardiovascular disease [2–4]. Epidemiologic data shows that fetuses and neonates with IUGR are predisposed to metabolic syndromes later in life [5]. Additionally, modern lifestyle factors, such as unhealthy diets, lack of physical activity, and chronic stress, exacerbate these IUGR-induced long-term metabolic issues, underscoring the urgency of addressing this significant public health concern [6].

The liver, at the core of metabolic issues, is a major target of IUGR [7,8]. For instance, the liver undergoes structural and functional changes in response to adverse intrauterine conditions [9]. Among the several damages in the liver, impaired fatty acid oxidation (FAO) and lipid decomposition are found to be at the core of metabolic issues caused by IUGR [9–11]. Studies on IUGR pigs have shown increased fatty acid flux toward the liver and reduced lipolysis and fatty acid oxidation [9]. Further, IUGR not only causes hypoxic stress in the early stage but also persistent, aberrant metabolism, particularly in lipid accumulation and hypoxia injury [9,12]. This underscores the critical need for early intervention and management to mitigate long-term live lipid disorders associated with IUGR.

Folic acid (FA) is often used as a dietary supplement during pregnancy and lactation to support fetal and infant development [13]. Previous studies have shown that a low dose of dietary FA supplementation (5 mg/kg) during pregnancy could improve IUGR by alleviating placental inflammation and oxidative stress [14]. It has been shown to ameliorate abnormal hepatic lipid metabolism and reduce hepatic lipid accumulation by upregulating peroxisome proliferator-activated receptor α (PPARα) levels in high-fat diet-induced fatty liver disease [15, 16]. Huang *et al.* have suggested

that maternal supplementation with high FA improves glucose intolerance and insulin resistance in male mice offspring fed a high-fat diet [17]. However, the effectiveness of FA for already-born IUGR neonates on lipid metabolism, particularly regarding its long-term effects into adulthood, remains uncertain and requires further research.

In the present study, we aim to explore the long-term effects of FA supplementation on lipid metabolism in IUGR offspring. Our findings indicate that IUGR induces long-term lipid metabolic dysfunction in offspring. Additionally, dietary administering FA in early post-natal life can ameliorate these metabolic alterations and gene expressions in the liver of IUGR rats, with a focus on the FAO process. This study highlights the potential of FA supplementation in managing IUGR-induced metabolic disorders and improving the long-term health outcomes of affected individuals.

## Methods

### Animal experiments

The experimental protocol was approved by the Laboratory Animal Center of the Third Xiangya Hospital, Central South University (Ethics number: CSU-2023–0244). A total of Sixteen 9-week-old Sprague-Dawley rats were selected, including 8 male and 8 female rats. The experimental animals were placed in a temperature-controlled room with a 12-hour light/dark cycle, and after 7 days of acclimation to the laboratory environment, the male and female rats were placed in a single cage in a 1:1 ratio, and the day when the vaginal plug appeared was designated as the first day of pregnancy.

The pregnant rats were randomly assigned to either the IUGR group or the control (CON) group using a random number generator. We induced IUGR models using maternal calorie and protein restriction, which has been previously validated [18]. In the IUGR group, pregnant rats were fed a 10% protein diet throughout gestation (protein-restricted diet), while those in the CON group received a 21% protein (normal) diet. Both groups received the 21% protein diet during lactation (S1 Table).

There was no difference in litter size between control and food-restricted pregnancies (15.8 ± 1.0 vs 16.0 ± 0.8 pups/litter). Offspring with a birth weight lower than two standard deviations from the mean of the CON group were classified as IUGR. The incidence rate of IUGR in control and IUGR groups was 1.6% and 85%, respectively. To investigate the effects of postnatal FA intervention, a subset of the offspring received a diet supplemented with FA, establishing four experimental groups for study: CON, CON+FA, IUGR, and IUGR+FA. Offspring in the IUGR+ folic acid (IUGR+FA) group and the CON+FA group were fed with folic acid 5 mg/kg for 3 weeks after weaning at 21 days, and then fed with a normal diet for 3 months (Also see S2A Fig). FA supplementation was initiated at weaning (postnatal day 21) to model an early postnatal intervention during a period of high developmental plasticity, and the timing and dose (5 mg/kg) were selected based on previous studies [14].

The rats were monitored at least twice daily by trained personnel for health status and behavioral changes. Parameters for assessment included, but were not limited to: general activity levels, food and water consumption, coat condition, and any signs of distress (such as vocalization, self-isolation, or aggressive behavior).

If rats exhibited any of the following irreversible and severe signs of distress that could not be alleviated through intervention, such as weight loss exceeding 20% of initial body weight, inability to eat or drink voluntarily, or the development of severe neurological symptoms or organ failure, euthanasia was performed immediately in accordance with the method described below. All other offspring were euthanized at the planned experimental endpoints: postnatal day (P) 1, P7, P21, P60, and P90. At these time points, offspring were administered a deep anesthetic overdose via intraperitoneal injection of pentobarbital sodium at a dose of 150 mg/kg. This ensured the animals rapidly reached a state of deep anesthesia and pain insensibility. Death was confirmed by the absence of corneal reflex, loss of response to toe/tail pinch, and cessation of respiration, followed by cervical dislocation as a secondary physical method of euthanasia. The liver index was calculated as (liver weight / body weight) × 100%. To minimize procedural stress, all interventions were performed in a quiet environment by personnel trained in humane animal handling techniques.

## Histological analysis

The liver tissues were embedded in paraffin and 5 µM sections. Paraffin-embedded sections were heated at 60°C for 2 hours and dewaxed by xylene, followed by serial rehydration steps (100% ethanol, 80% ethanol, 50% ethanol; each 5 min). Then, sections were stained with hematoxylin and eosin using a standard staining technique. Finally, images were captured on a fluorescent microscope (Olympus BX5, Tokyo, Japan).

## ELISA

Venous blood samples for rats were drawn and stored in the ethylene diamine tetraacetate acid (EDTA) containing tubes, all the blood samples were centrifuged for 15 minutes at 3000 rpm to gain the plasma. The plasma concentration of free fatty acid (A042-2–1, Nanjing Institute of Biological Engineering) and triglyceride (F001-2, Nanjing Institute of Biological Engineering) were measured respectively according to the manufacturer's instructions.

## Quantitative real-time polymerase chain reaction

The total RNA of the liver was extracted with Trizol reagent (Invitrogen, Shanghai, China), and was reverse transcribed using Superscript II reverse transcriptase kits (both from Invitrogen). Diluted cDNAs were used as templates for qPCR with TAKARA gene expression assays (TAKARA Biotechnology Co., LTD. Dalian, China) and PCR mastermix in ABI7300 thermal cycler according to the manufacturer's instructions (Applied Biosystems, Foster City, CA). The primer sequences used were shown as follows:

| Gene | Sense (5'-3') | Anti-sense (5'-3') |
|---|---|---|
| β-actin | ACATCCGTAAAGACCTCTATGCC | TACTCCTGCTTGCTGATCCAC |
| Acox1 | ACTATATTTGGCCAATTTTGTGGA | TCGAAGATGAGTTCCGTGGC |
| Acox3 | AACCTGCTCCACCCTCAGA | TCCTAGTACTGGCGTCGGTC |
| CPT1 | GCAGCTCGCACATTACAAGG | CTCTGTCCTCCCTTCTCGGA |
| PPARα | GGCTCTGAACATTGGCGTTC | CAAGGGGACAACCAGAGGAC |

## Western blot analysis

Livers were homogenized in proteinase inhibitor-containing RIPA lysis solution before being centrifuged at 4 °C for 15 min, and the supernatants were collected. Each sample, containing 20 µg of protein, was boiled in the loading buffer before being transferred to 10% SDS-polyacrylamide gels for electrophoresis and then added to a PVDF membrane.

For blocking 5% dry non-fat milk dissolved in Tris-buffered saline with 0.05% Tween 20 as detergent (TBST) was used. Membranes were then incubated with antibodies (β-actin,1:5000; Acox1,1:1000; Acox3,1:1000; CPT1, 1:1000; PPARα,1:500) at 4 °C overnight, followed by peroxidase-conjugated secondary antibody, or incubated with HRP-conjugated beta-actin monoclonal antibody. Blots were detected by an imaging system (ChemiScope6100, Shanghai, China).

## Statistical analysis

The test data were initially processed by SPSS22.0 statistical software (Point Richmond, CA). A Kolmogorov's-Smirnov test was used to test the normality of the data, and all normally distributed continuous variables were described as mean and standard deviation. For comparisons between 2 groups, statistical significance was calculated using Student's *t*-test. Differences between the 4 groups were evaluated using one-way ANOVA. For longitudinal data, a repeated-measures ANOVA or mixed-effects model was employed to account for within-subject correlations over time. * $P < 0.05$ was considered statistically significant.

## Results

### IUGR induces persistent lipid metabolism dysfunction in offspring

A total of 146 rats were used in this experiment. 141 of them were euthanized and 5 died during the perinatal period. As previously reported, the body weights of the IUGR group were significantly lower compared to the normal group from birth to 21 days (S1 Fig B). After 3 weeks of age, IUGR offspring exhibited catch-up growth, and the differences in body weight were normalized at 60 days after birth (S1 Fig B).

Given that the liver is the principal target organ in response to adverse intrauterine conditions, we then examine the effect of IUGR on the liver. Our results demonstrated that liver weight and liver index levels are decreased in infant IUGR rats compared to control rats, while no significant differences are observed during adolescence (S1 Fig C-D). Consistent with previous studies [19], levels of serum triglycerides (TG) and free fatty acids (FFA) significantly increased in IUGR rats compared to control rats during the early stages. Furthermore, we found that these lipid alterations persisted into adulthood in IUGR rats, at the 90-day age the TG and FFA were significantly higher when compared to controls (TG: 2.9-fold; FFA: 4.1-fold) (S1 Fig E-F). Compared to the normal structure in control rats, H&E staining exhibited enlarged hepatocytes, intercellular spaces, and extensive hepatocellular vacuoles in the liver of IUGR rats at both 1 day and 90 days after birth, indicative of the abnormalities in hepatic lipid metabolism (S1 Fig G).

At 90 days after birth, IUGR downregulated the expression of lipid oxidation-related genes, including PPARα, as well as Carnitine palmitoyl transferase 1 (Cpt1), acyl-CoA oxidase 1 (Acox1) and Acox3, which are known to be components in the regulatory network of fatty acid β-oxidation (PPARα: 0.12-fold; Ctp1:0.10-fold; Acox1:0.18-fold; Acox3: 0.11-fold) (S1 Fig H). These results suggest the IUGR-induced deficiency of the PPARα network and dysfunction of lipid metabolism persists into adulthood.

### The impact of early postnatal folic acid supplementation on IUGR-induced lipid metabolic dysfunction in adulthood

To determine if folic acid supplementation in the early postpartum period could counteract IUGR-induced metabolic dysfunction, offspring were fed either a control or folic acid-supplemented diet for 3 weeks after weaning (S2 Fig A). Folic acid intervention did not affect the general appearance or weight of both IUGR and control rats (S2 Fig B). However, folic acid addition acutely reduced serum TG and FFA levels in adult IUGR rats at 60 days and 90 days of age (S2 Fig C-D). The IUGR-induced hepatic changes were reversed by folic acid supplementation. Compared to livers from the IUGR group, livers from the IUGR+FA group showed clear tissue structure and less vacuolization, as revealed by H&E staining (S2 Fig E). These results suggest that folic acid supplementation can mitigate IUGR-induced lipid metabolic dysfunction and hepatic abnormalities in adulthood.

### Folic acid upregulates lipid-related genes in IUGR Rats

Then we determined the effect of folic acid on lipid-related genes in the liver. As shown in S3 Fig A-D, early postnatal folic acid supplementation significantly upregulated the expression of lipid-related genes in the liver from adult IUGR rats, including PPARα, CPT1, Acox1 and Acox3, while not affecting control rats (S3 Fig A-D). Correspondingly, the expression levels of these proteins were also significantly upregulated after folic acid treatment (S3 Fig E-I). Furthermore, to explore a potential epigenetic mechanism, we assessed global DNA methylation levels in the liver. IUGR was associated with significantly elevated DNA methylation at 90 days. Postnatal FA supplementation effectively reduced this IUGR-induced hypermethylation in the IUGR+FA group (S3 Fig J, K). Overall, folic acid improves hepatic lipid metabolism by upregulating PPARα levels and β-oxidation related genes.

 

## Discussion

In the present study, we confirm that newborns with IUGR individuals are predisposed to lipid metabolism abnormalities in adulthood. Focused on the persistent impact of IUGR on metabolic health, we found that early postnatal FA supplementation may have potential ameliorative effects on metabolic dysfunction by increasing the level of PPARα and related genes (CPT1, ACOX1, ACOX3) in the liver. This suggests that FA administration during early life can counteract the long-term effects of IUGR on lipid metabolism in adulthood.

The concept of "developmental origins of adult health and disease (DOHaD)" proposed by Barker, posits that IUGR shapes a "thrifty phenotype" through intrauterine reprogramming, predisposing individuals to metabolic syndrome in adulthood [20, 21]. Epidemiological and experimental research has reported that IUGR individuals are susceptible to metabolic syndrome and related disorders, such as obesity, diabetes, and cardiovascular diseases in later life [22]. In this context, we have established a model of maternal food restriction during pregnancy that results in IUGR newborns. After being provided with a standard postweaning diet, these IUGR newborns demonstrate rapid catch-up growth. Our results confirm that IUGR induces persistent lipid metabolism dysfunction in offspring, as evidenced by increased TG and FFA levels, even in the absence of significant differences in body weight. While efforts have traditionally focused on improving the intrauterine environment to prevent IUGR, it is equally important to address the needs of those already born with IUGR, who belong to a high-risk population [23]. These individuals are particularly susceptible to environmental influences, such as modern lifestyles and unhealthy dietary habits, which can lead to the development of metabolic syndrome, worse quality of life, and increases in healthcare burdens [8].

Our research also observed that hepatic abnormalities were evident as early as one day after birth, with pathological changes including enlarged cells, irregular arrangement, loose intercellular spaces and vacuolar degeneration. Furthermore, we found the downregulation of genes related to the fatty acid β-oxidation process, suggesting that the liver of adult IUGR rats is unable to efficiently transport, oxidize, and utilize lipids. Recent studies have found that oxidative stress-induced liver damage from IUGR appears in infancy and accumulates with age, becoming a core factor in metabolic abnormalities in adulthood [9,10]. These findings emphasize that the liver, a crucial metabolic organ, is highly vulnerable to IUGR. Additionally, the liver's high plasticity and responsiveness to nutritional interventions in the postnatal period suggest a critical window for addressing liver injury.

FA is commonly used as dietary supplements during pregnancy and lactation to support optimal birth and nurturing [24]. The beneficial effects of FA on lipid metabolism have been documented in high-fat diet-induced fatty liver disease [25, 26]. Previous studies also reported that maternal supplementation with FA during pregnancy and lactation improves glucose intolerance and insulin resistance in offspring fed a high-fat diet [27]. In this study, we extended its application into early postnatal life for offspring in our study. We found that post-weaning FA supplementation for newborns can also reduce the lipid level and hepatic injury induced by IUGR. The significant long-term impact of post-weaning FA supplementation provides an early treatment option for IUGR populations at high risk of metabolic diseases.

The mechanisms by which FA regulates lipid metabolism in the context of IUGR are not fully understood, involving mitigating oxidative stress [28], DNA methylation [29], inflammation [30], and so on. As reported, PPARα signals were essential for regulating TG deposition and hepatic damage for IUGR [14,19]. Our findings indicate that FA upregulates the expression of PPARα and its target genes involved in fatty acid oxidation. The mRNA and protein level of PPARα, CPT1, Acox1 and Acox3 was observed to be reduced by IUGR but were restored with FA intervention. As we know, PPARα, abundant in the liver, is a pivotal transcriptional regulator of lipid metabolism [15]. The genes CPT1, Acox1 and Acox3, regulated by PPARα, are involved in fatty acid uptake and activation and mitochondrial and peroxisomal fatty acid oxidation [25]. Notably, the PPARα signaling pathway is dysregulated in IUGR mice [9,12,19, 31]. In addition, PPARα activation can mitigate hepatic damage in IUGR mice, suggesting that PPARα could be a potential therapeutic target for treating hepatic dysfunction associated with IUGR [9]. Based on these observations, we hypothesize that PPARα is one of the important mechanisms through which FA regulates lipid metabolism in IUGR. Further research is needed to fully elucidate

the underlying mechanisms of FA's regulation of the PPARα pathway and to assess the long-term outcomes of such interventions.

In conclusion, our study demonstrates that IUGR induces long-term liver injury and lipid metabolic dysfunction in offspring, which can be ameliorated by early postnatal FA supplementation. We propose that PPARα is not only a key signaling mechanism in IUGR-induced lipid metabolic abnormalities but also a crucial target through which FA exerts its effects. Taken together, these data enhance our understanding of the metabolic characteristics of the liver in IUGR and propose potential therapeutic targets for IUGR-induced long-term metabolic diseases.

## Supporting information

**S1 Fig. IUGR induces lipid metabolism dysfunction that persists into adulthood in offspring.**
(PDF)

**S2 Fig. The impact of early postnatal folic acid supplementation on IUGR-induced lipid metabolic dysfunction in adulthood.**
(PDF)

**S3 Fig. Folic acid upregulates lipid-related genes in IUGR Rats.**
(PDF)

**S1 Table. Composition and nutrient composition of experimental feed.**
(PDF)

**S1 Data. Raw data.**
(ZIP)

## Author contributions

**Conceptualization:** Laiyi Zhou Weiyun Shen.

**Data curation:** Laiyi Zhou, Weiyun Shen.

**Formal analysis:** Laiyi Zhou, Weiyun Shen.

**Investigation:** Laiyi Zhou, Weiyun Shen.

**Methodology:** Laiyi Zhou, Weiyun Shen.

**Writing – original draft:** Laiyi Zhou, Weiyun Shen, Xiaori He.

**Writing – review & editing:** Can Liang, Qingyi Dong, Jingwei Wang.

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
