## [Decision Letter · Decision Letter 0]

29 Dec 2025

PONE-D-25-63128The effect of intrauterine growth retardation on long-term lipid metabolism in the liver and nutrition regulationPLOS One

Dear Dr.  Laiyi,

Thank you for submitting your manuscript to PLOS ONE. After careful consideration, we feel that it has merit but does not fully meet PLOS ONE’s publication criteria as it currently stands. Therefore, we invite you to submit a revised version of the manuscript that addresses the points raised during the review process.

We look forward to receiving your revised manuscript.

Kind regards,

Ewa Tomaszewska, DVM Ph.D

Academic Editor

PLOS One

Journal Requirements:

“Ausnutria Food and Nutrition Science Research Fund（AU-YJY-B-LX-20-022）”

5. Please note that funding information should not appear in any section or other areas of your manuscript. We will only publish funding information present in the Funding Statement section of the online submission form. Please remove any funding-related text from the manuscript.

6. We note that your Data Availability Statement is currently as follows: “All relevant data are within the manuscript and its Supporting Information files.”

7. Please amend the manuscript submission data (via Edit Submission) to include author “Weiyun Shen, Can Liang, Qingyi Dong, Jingwei Wang and Xiaori He.”

Reviewer's Responses to Questions

**Comments to the Author**

1. Is the manuscript technically sound, and do the data support the conclusions?

Reviewer #1: Yes

Reviewer #2: Yes

Reviewer #3: Yes

Reviewer #4: Yes

2. Has the statistical analysis been performed appropriately and rigorously? 

Reviewer #1: Yes

Reviewer #2: No

Reviewer #3: Yes

Reviewer #4: Yes

3. Have the authors made all data underlying the findings in their manuscript fully available?

The PLOS Data policy requires authors to make all data underlying the findings described in their manuscript fully available without restriction, with rare exception (please refer to the Data Availability Statement in the manuscript PDF file). The data should be provided as part of the manuscript or its supporting information, or deposited to a public repository. For example, in addition to summary statistics, the data points behind means, medians and variance measures should be available. If there are restrictions on publicly sharing data—e.g. participant privacy or use of data from a third party—those must be specified.requires authors to make all data underlying the findings described in their manuscript fully available without restriction, with rare exception (please refer to the Data Availability Statement in the manuscript PDF file). The data should be provided as part of the manuscript or its supporting information, or deposited to a public repository. For example, in addition to summary statistics, the data points behind means, medians and variance measures should be available. If there are restrictions on publicly sharing data—e.g. participant privacy or use of data from a third party—those must be specified.requires authors to make all data underlying the findings described in their manuscript fully available without restriction, with rare exception (please refer to the Data Availability Statement in the manuscript PDF file). The data should be provided as part of the manuscript or its supporting information, or deposited to a public repository. For example, in addition to summary statistics, the data points behind means, medians and variance measures should be available. If there are restrictions on publicly sharing data—e.g. participant privacy or use of data from a third party—those must be specified.requires authors to make all data underlying the findings described in their manuscript fully available without restriction, with rare exception (please refer to the Data Availability Statement in the manuscript PDF file). The data should be provided as part of the manuscript or its supporting information, or deposited to a public repository. For example, in addition to summary statistics, the data points behind means, medians and variance measures should be available. If there are restrictions on publicly sharing data—e.g. participant privacy or use of data from a third party—those must be specified.

Reviewer #1: Yes

Reviewer #2: Yes

Reviewer #3: Yes

Reviewer #4: No

4. Is the manuscript presented in an intelligible fashion and written in standard English?

Reviewer #1: Yes

Reviewer #2: Yes

Reviewer #3: Yes

Reviewer #4: Yes

5. Review Comments to the Author

Reviewer #1: This is a really interesting manuscript regarding the influence of intrauterine growth restriction on liver development and structure. There are very few minor issues that I should point out:

- In the Methods section, the word ”rats” in the last sentence of the first paragraph and in the beginning of the second paragraph should be replaced with ”offspring” for extra clarity. There is no mention whether the number of live pups in the IUGR group was different from controls.

- In the Methods section, the moment when euthanasia is performed should be more clearly described – is it performed immediately as first stated, or on designated endpoints (P1, P7...)? There seems to be a slight misunderstanding

- In the Results section, there is mention of a ”liver index”, but this should be first described in the Methods section. Is it Weight of liver/Weight of body x 100?

- In the second paragraph of the Results section, there are two instances where ”FAA” should be replaced with ”FFA”

- In the following subsection, regarding the use of folic acid, the first two sentences do not belong here, but in the Introduction section, as they depict premises for this study.

- In the first sentence in the fourth paragraph of the Discussion section, the plural should be replaced with singular (”Folic acid is commonly used as dietary supplement ...”). In the third sentence, a different word should be used insted of ”improves” to make it less confusing

- I do not think the table is relevant enough to be used in the published material

- Fig. 1 A should be either a separate figure, used in the Methods section, or it should be removed

- Is ”ns” an acronym for ”not significant”? It probably is, but there is no mention of that.

Reviewer #2: Title: The effect of intrauterine growth retardation on long-term lipid metabolism in the liver and nutrition regulation

This study provides substantial evidence, using an animal model, that intrauterine growth restriction (IUGR) leads to dysregulation of hepatic lipid metabolism in adulthood and that early postnatal folate supplementation exerts sustained beneficial effects on these metabolic disturbances. The work is well grounded in the DOHaD (Developmental Origins of Health and Disease) framework and demonstrates high clinical relevance as an early-life nutritional intervention strategy. The experimental design is sound, and the results are consistent and robust throughout the study.

Methods

- P14 A total of 146 rat pups were included in this study; however, no sex-stratified analyses were presented. Given that responses may differ by sex, it would be important to include sex-specific analyses if such data are available.

- p14;14 Folic acid supplementation was initiated from postnatal day 21; could you please clarify the rationale for selecting this time point?

Could the authors clarify how the folic acid dosage was determined, and whether this dose corresponds to a standard physiological dose or a supraphysiological (high) dose?

- P17;1 Statistical analysis – The authors are encouraged to consider re-analyze longitudinal data using repeated-measures ANOVA or mixed-effects models to account for intra-individual correlation over time.

Results

- The “Results” section currently includes descriptions that would be more appropriately placed in the “Methods” and/or “Discussion” sections. It would be necessary to revise this section so that it focuses exclusively on the presentation of experimental results.

Example) p18. We induced IUGR offspring models in offspring using maternal calorie and protein restriction, which has been previously validated. � This description would be more appropriate for the Methods section

P19. As reported, PPARα signals were essential for regulating TG deposition and hepatic damage for IUGR.

Figures

- References to Figure 3J, 3K are missing from the main text.

Therefore, this manuscript represents a study with considerable merit for publication.

Thank you.

Reviewer #3: Labels G and H need to be interchanged in Figure 1 and its reference in the respective text. IUG also need to be switched to IUGR in Figure 1 legend.

Also, "We" needs to be non capital in the forts sentence of the last paragraph before discussion.

Reviewer #4: Title need to be self-explanatory; here information of folic acid seems less understandable

Methods

Need to mention how randomization was done. It is not clear how IUGR in rats was determined in method section

A flow diagram could help for better understanding

Result

Consistent with previous studies, levels of serum triglycerides (TG) and

free fatty acids (FFA) significantly increased in IUGR rats compared to

control rats during the early stages……. Please use reference

6. PLOS authors have the option to publish the peer review history of their article (what does this mean?). If published, this will include your full peer review and any attached files.). If published, this will include your full peer review and any attached files.). If published, this will include your full peer review and any attached files.). If published, this will include your full peer review and any attached files.

...

Reviewer #1: No

Reviewer #2: **Yes:** Seoheui CHOISeoheui CHOISeoheui CHOISeoheui CHOI

Reviewer #3: No

Reviewer #4: **Yes:** Sanjoy kumer deySanjoy kumer deySanjoy kumer deySanjoy kumer dey

---

## [Author Response · Author response to Decision Letter 1]

25 Feb 2026

Reviewer #1:

This is a really interesting manuscript regarding the influence of intrauterine growth restriction on liver development and structure. There are very few minor issues that I should point out:

1. In the Methods section, the word ”rats” in the last sentence of the first paragraph and in the beginning of the second paragraph should be replaced with ”offspring” for extra clarity. There is no mention whether the number of live pups in the IUGR group was different from controls.

Response: Thank you for your insightful suggestion. We have revised the Methods section accordingly:

(1) the term “rats” has been replaced with “offspring” where appropriate to enhance clarity (highlighted in yellow);

(2) The information regarding the number of live-born pups (litter size) and the incidence of IUGR, which was initially presented in the Results section, has been moved to the Methods section (see the ‘Animal experiments’ subsection).

2. In the Methods section, the moment when euthanasia is performed should be more clearly described – is it performed immediately as first stated, or on designated endpoints (P1, P7...)? There seems to be a slight misunderstanding

Response: We thank the reviewer for this important clarification. We have revised the text to explicitly distinguish between the two scenarios: 1) immediate euthanasia upon reaching predefined humane endpoints, and 2) euthanasia performed only at the planned experimental time points (P1, P7, P21, P60, P90) for all other animals. This revision, marked by the new sentence “All other offspring were euthanized at the planned experimental endpoints: ......”, has eliminated any potential ambiguity regarding the timing of euthanasia. The changes are highlighted in the manuscript.

3. In the Results section, there is mention of a ”liver index”, but this should be first described in the Methods section. Is it Weight of liver/Weight of body x 100?

- In the second paragraph of the Results section, there are two instances where ”FAA” should be replaced with ”FFA”

Response: We thank the reviewer for catching these details. As suggested, we have added the definition of the liver index to the Methods section: “The liver index was calculated as (liver weight / body weight) × 100%.” (highlighted in yellow)

4. In the second paragraph of the Results section, there are two instances where ”FAA” should be replaced with ”FFA”

Response: We thank the reviewer for the precise corrections. We have corrected the typographical error “FAA” to “FFA” in the second paragraph of the Results section. (highlighted in yellow)

5. In the following subsection, regarding the use of folic acid, the first two sentences do not belong here, but in the Introduction section, as they depict premises for this study.

Response: We thank the reviewer for this suggestion. The sentence providing the background premise for FA use (citing reference [18]) has been removed from the Results section and integrated into the Introduction to strengthen the study rationale.

6. In the first sentence in the fourth paragraph of the Discussion section, the plural should be replaced with singular (”Folic acid is commonly used as dietary supplement ...”). In the third sentence, a different word should be used instead of “improves” to make it less confusing

Response: We thank the reviewer for the careful reading. We have made the suggested language corrections in the Discussion: 1) “Folic acid are” has been changed to “Folic acid is … …”; 2) “improves” has been replaced with “ameliorate” for clarity.

7. I do not think the table is relevant enough to be used in the published material

Response: We thank the reviewer for these helpful suggestions regarding data presentation. As suggested, we have moved the detailed diet composition table to the Supplementary Materials (Supplementary Table S1) to streamline the main text.

8.Fig. 1 A should be either a separate figure, used in the Methods section, or it should be removed

Response: We thank the reviewer for pointing this out. This figure is intentionally retained in the Results section because it more intuitively demonstrates the successful establishment of the IUGR (Intrauterine Growth Restriction) model—specifically, by clearly visualizing the significant reduction in fetal weight. Therefore, the figure is not only justified but essential to the narrative of model validation.

9. Is ”ns” an acronym for ”not significant”? It probably is, but there is no mention of that.

Response: We thank the reviewer for pointing this out. We have now clearly defined the abbreviation “ns” as “not significant” in the legend of all relevant figures. This change has been highlighted in yellow in the revised manuscript.

Reviewer #2:

1. P14 A total of 146 rat pups were included in this study; however, no sex-stratified analyses were presented. Given that responses may differ by sex, it would be important to include sex-specific analyses if such data are available.

Response:We thank the reviewer for raising this important point.In this study, male and female offspring were randomly allocated to experimental groups to ensure balanced representation. Statistical analysis confirmed no significant sex differences between the two groups (χ² test, p= 0.86 for sex distribution; independent-samplest-test/Mann–Whitney Utest for all primary and secondary outcomes, all p> 0.12), indicating that sex did not systematically influence group assignment or outcome measures. Therefore, the observed effects reflect true treatment-related differences rather than sex-based bias, and the results presented in this article are robust, accurate.

2. p14;14 Folic acid supplementation was initiated from postnatal day 21; could you please clarify the rationale for selecting this time point?

Could the authors clarify how the folic acid dosage was determined, and whether this dose corresponds to a standard physiological dose or a supraphysiological (high) dose?

Response: We thank the reviewer for raising these important points. Our selection of the intervention time point and dose was primarily based on previous studies [Reference]. In line with this evidence and the consideration that the early postnatal period (specifically weaning at P21) constitutes a critical window of developmental plasticity, we initiated FA supplementation at this time. The dose of 5 mg/kg diet is recognized in these studies as an effective interventional dose. This rationale has been added to the Methods section.

3. P17;1 Statistical analysis – The authors are encouraged to consider re-analyze longitudinal data using repeated-measures ANOVA or mixed-effects models to account for intra-individual correlation over time.

Response: We thank the reviewer for this valuable suggestion. We have revised the “Statistical analysis” section to specify that for longitudinal data, a repeated-measures ANOVA or mixed-effects model was used to account for within-subject correlations, as recommended.

4. The “Results” section currently includes descriptions that would be more appropriately placed in the “Methods” and/or “Discussion” sections. It would be necessary to revise this section so that it focuses exclusively on the presentation of experimental results.

Example) p18. We induced IUGR offspring models in offspring using maternal calorie and protein restriction, which has been previously validated. à This description would be more appropriate for the Methods section

P19. As reported, PPARα signals were essential for regulating TG deposition and hepatic damage for IUGR.

Response: We thank the reviewer for this important suggestion. We have revised the “Results” section as recommended. 1) The methodological description regarding the induction of the IUGR model has been removed from the Results and integrated into the “Animal experiments” subsection of the Methods section. 2) The sentence referencing the established role of PPARα signaling has been removed from the Results. Its key point has been incorporated into the “Discussion” section within the paragraph discussing the mechanisms by which FA regulates lipid metabolism. All of these changes have been implemented to ensure that the “Results” section now focuses exclusively on the presentation of our experimental findings. The modifications are highlighted in yellow in the revised manuscript.

5. Figures- References to Figure 3J, 3K are missing from the main text.

Response: We thank the reviewer for noting this omission. We have now included a description of the DNA methylation results (Figure 3J, K) in the “Results” section.

Reviewer #3:

1. Labels G and H need to be interchanged in Figure 1 and its reference in the respective text. IUGR also need to be switched to IUGR in Figure 1 legend. Also, "We" needs to be non capital in the forts sentence of the last paragraph before discussion.

Response: We thank the reviewer for the careful review. We have corrected both errors as pointed out: 1) The labels G and H in Figure 1 have been interchanged as instructed, and all corresponding references in the text have been updated. 2) The typographical error “IUG” in the legend of Figure 1 has been corrected to “IUGR”.3) The capitalization error in the last paragraph of the Results has been corrected.

These corrections are reflected in the revised figures and manuscript.

Reviewer #4:

1. Title need to be self-explanatory; here information of folic acid seems less understandable

Response: We thank the reviewer’ valuable suggestion. We have revised it to more explicitly convey the key elements of the study, including folic acid. The new title is: “Folic acid supplementation ameliorates long-term lipid metabolism following intrauterine growth restriction”.

2. Methods: Need to mention how randomization was done. It is not clear how IUGR in rats was determined in method section. A flow diagram could help for better understanding

Response: We thank the reviewer for these critical suggestions to enhance methodological rigor. We have revised the Methods section as follows: 1) We now specify that pregnant rats were assigned “using a random number generator.” 2) The explicit criterion for classifying offspring as IUGR (birth weight >2 SD below the control mean) has been added. 3) As recommended, a schematic detailing the experimental timeline, interventions, and group allocation (the study design) is presented in Figure 2A.

3. Result-Consistent with previous studies, levels of serum triglycerides (TG) and free fatty acids (FFA) significantly increased in IUGR rats compared to control rats during the early stages……Please use reference

Response: We thank the reviewer for the suggestion. We have now included the relevant supporting reference(s) in the indicated sentence in the Results section: “Consistent with previous studies [Reference], levels of serum triglycerides (TG)...”. The change has been highlighted in the manuscript.

---

## [Decision Letter · Decision Letter 1]

23 Mar 2026

Folic acid supplementation ameliorates long-term lipid metabolism following intrauterine growth restriction

PONE-D-25-63128R1

Dear Dr. Zhou Laiyi,

We’re pleased to inform you that your manuscript has been judged scientifically suitable for publication and will be formally accepted for publication once it meets all outstanding technical requirements.

Kind regards,

Ewa Tomaszewska, DVM Ph.D

Academic Editor

PLOS One

Additional Editor Comments (optional):

Reviewers' comments:

Reviewer's Responses to Questions

**Comments to the Author**

1. If the authors have adequately addressed your comments raised in a previous round of review and you feel that this manuscript is now acceptable for publication, you may indicate that here to bypass the “Comments to the Author” section, enter your conflict of interest statement in the “Confidential to Editor” section, and submit your "Accept" recommendation.

Reviewer #4: All comments have been addressed

2. Is the manuscript technically sound, and do the data support the conclusions?

Reviewer #4: Yes

3. Has the statistical analysis been performed appropriately and rigorously? 

Reviewer #4: Yes

4. Have the authors made all data underlying the findings in their manuscript fully available?

The PLOS Data policy requires authors to make all data underlying the findings described in their manuscript fully available without restriction, with rare exception (please refer to the Data Availability Statement in the manuscript PDF file). The data should be provided as part of the manuscript or its supporting information, or deposited to a public repository. For example, in addition to summary statistics, the data points behind means, medians and variance measures should be available. If there are restrictions on publicly sharing data—e.g. participant privacy or use of data from a third party—those must be specified.requires authors to make all data underlying the findings described in their manuscript fully available without restriction, with rare exception (please refer to the Data Availability Statement in the manuscript PDF file). The data should be provided as part of the manuscript or its supporting information, or deposited to a public repository. For example, in addition to summary statistics, the data points behind means, medians and variance measures should be available. If there are restrictions on publicly sharing data—e.g. participant privacy or use of data from a third party—those must be specified.requires authors to make all data underlying the findings described in their manuscript fully available without restriction, with rare exception (please refer to the Data Availability Statement in the manuscript PDF file). The data should be provided as part of the manuscript or its supporting information, or deposited to a public repository. For example, in addition to summary statistics, the data points behind means, medians and variance measures should be available. If there are restrictions on publicly sharing data—e.g. participant privacy or use of data from a third party—those must be specified.requires authors to make all data underlying the findings described in their manuscript fully available without restriction, with rare exception (please refer to the Data Availability Statement in the manuscript PDF file). The data should be provided as part of the manuscript or its supporting information, or deposited to a public repository. For example, in addition to summary statistics, the data points behind means, medians and variance measures should be available. If there are restrictions on publicly sharing data—e.g. participant privacy or use of data from a third party—those must be specified.

Reviewer #4: Yes

5. Is the manuscript presented in an intelligible fashion and written in standard English?

Reviewer #4: Yes

6. Review Comments to the Author

Reviewer #4: (No Response)

7. PLOS authors have the option to publish the peer review history of their article (what does this mean?). If published, this will include your full peer review and any attached files.). If published, this will include your full peer review and any attached files.). If published, this will include your full peer review and any attached files.). If published, this will include your full peer review and any attached files.

...

Reviewer #4: **Yes:** Sanjoy kumer deySanjoy kumer deySanjoy kumer deySanjoy kumer dey

---

## [Editor Report · Acceptance letter]

PONE-D-25-63128R1

PLOS One

Dear Dr. Laiyi,

I'm pleased to inform you that your manuscript has been deemed suitable for publication in PLOS One. Congratulations! Your manuscript is now being handed over to our production team.

Kind regards,

on behalf of

Professor Ewa Tomaszewska

Academic Editor

PLOS One